# Automated genome mining predicts structural diversity and taxonomic distribution of peptide metallophores across bacteria

Zachary L Reitz[1,2†], Bita Pourmohsenin[3†], Melanie Susman[4], Emil Thomsen[4], Daniel Roth[4], Alison Butler[4], Nadine Ziemert[3]*, Marnix H Medema[1]*

[1]Bioinformatics Group, Wageningen University, Wageningen, Netherlands; [2]Department of Evolution, Ecology, and Marine Biology, University of California, Santa Barbara, Santa Barbara, United States; [3]Interfaculty Institute of Microbiology and Infection Medicine, Institute for Bioinformatics and Medical Informatics, Tübingen, Germany; [4]Department of Chemistry and Biochemistry, University of California, Santa Barbara, Santa Barbara, United States

*For correspondence:
nadine.ziemert@uni-tuebingen.de (NZ);
marnix.medema@wur.nl (MHM)

†These authors contributed equally to this work

## eLife Assessment

This **important** and **compelling** study establishes a robust computational and experimental framework for the large-scale identification of metallophore biosynthetic clusters. The work advances beyond current standards, providing theoretical and practical value across microbiology, bioinformatics, and evolutionary biology.

**Abstract** Microbial competition for trace metals shapes their communities and interactions with humans and plants. Many bacteria scavenge trace metals with metallophores, small molecules that chelate environmental metal ions. Metallophore production may be predicted by genome mining, where genomes are scanned for homologs of known biosynthetic gene clusters (BGCs). However, accurately detecting non-ribosomal peptide (NRP) metallophore biosynthesis requires expert manual inspection, stymieing large-scale investigations. Here, we introduce automated identification of NRP metallophore BGCs through a comprehensive algorithm, implemented in antiSMASH, that detects chelator biosynthesis genes with 97% precision and 78% recall against manual curation. We showcase the utility of the detection algorithm by experimentally characterizing metallophores from several taxa. High-throughput NRP metallophore BGC detection enabled metallophore detection across 69,929 genomes spanning the bacterial kingdom. We predict that 25% of all bacterial non-ribosomal peptide synthetases encode metallophore production and that significant chemical diversity remains undiscovered. A reconstructed evolutionary history of NRP metallophores supports that some chelating groups may predate the Great Oxygenation Event. The inclusion of NRP metallophore detection in antiSMASH will aid non-expert researchers and continue to facilitate large-scale investigations into metallophore biology.

## Introduction

Across environments, microbes compete for a scarce pool of trace metals. Many microbes scavenge metal ions with small-molecule chelators called *metallophores*, which diffuse through the environment

and chelate metal ions with high affinity (*Hider and Kong, 2010*; *Kraemer et al., 2015*). A microbe possessing the right membrane transporters will be able to recognize and import a metallophore–metal complex, while other strains are unable to access the chelated metal ions. Thus, the metallophore secreted by one microbe can either support or inhibit growth of a neighboring strain, driving complex community dynamics in marine, freshwater, soil, and host environments (*Kramer et al., 2020*). The most well studied metallophores are the Fe(III)-binding *siderophores*, which have found applications in biocontrol, bioremediation, and medicine (*Soares, 2022*). Two recent studies demonstrated that the disease suppression ability of a rhizosphere microbiome is strongly determined by whether or not the pathogen can use siderophores produced by the community; a microbiome can even encourage pathogen growth when a compatible siderophore is produced (*Gu et al., 2020a*; *Gu et al., 2020b*). Compared to siderophores, other metallophore classes are relatively understudied, but they likely play equally important biological roles, as exemplified by recent reports of both commensal and pathogenic bacteria relying on zincophores to effectively colonize human hosts (*Behnsen et al., 2021*; *Mehdiratta et al., 2022*).

Hundreds of unique metallophore structures have been characterized, each with specific chemical properties (e.g., effective pH range, hydrophobicity, and metal selectivity) and biological effects on other microbes (based on membrane transporter compatibility). Experimentally characterizing metallophores can be time-consuming and costly, and thus researchers often use genome mining to predict metallophore production in silico (*Reitz and Medema, 2022*). Taxonomy alone is not sufficient to predict what metallophores will be produced by a microbe, as production can vary significantly even within a single species (*Cézard et al., 2015*). Instead, metallophores must be predicted from each genome based on the presence of biosynthetic gene clusters (BGCs) that encode their biosynthesis. The majority of known metallophores are non-ribosomal peptides (NRPs), a broad class of natural products that also includes many antibiotics, antitumor compounds, and toxins. Specialized chelating moieties bind directly to the metal ion (in the case of siderophores, $Fe^{3+}$), while other amino acids in the peptide chain give the metallophore the required flexibility for chelation. Nearly all NRP metallophores contain one or more of the substructures shown in *Figure 1A*: 2,3-dihydroxybenzoate (catechol, 2,3-DHB), hydroxamates, salicylate, β-hydroxyaspartate (β-OHAsp), β-hydroxyhistidine (β-OHHis), graminine, Dmaq (1,1-dimethyl-3-amino-1,2,3,4- tetrahydro-7,8-dihydroxy-quinoline), and the pyoverdine chromophore. Biosynthetic pathways are known for each of the chelating groups (*Figure 1B*), and the presence of a chelator pathway may be used as a marker for metallophore production.

Mining genomes for metallophore BGCs has facilitated the discovery of chemically and biologically diverse metallophore systems; however, automated detection tools are still severely lacking (*Reitz and Medema, 2022*) The peptidic backbones of NRP metallophores are produced by non-ribosomal peptide synthetases (NRPSs), large multi-domain enzymes that activate and condense amino acids and other substrates in an assembly-line manner (*Süssmuth and Mainz, 2017*). In the past two decades, a variety of bioinformatic tools have been developed to identify NRPS BGCs in a genome. One of the most popular is the secondary metabolite prediction platform antiSMASH, which uses a library of profile hidden Markov models (pHMMs) to identify (combinations of) enzyme-coding genes that are indicative of certain classes of specialized metabolite biosynthetic pathways (*Blin et al., 2021b*; *Blin et al., 2023*). For example, antiSMASH identifies an NRPS BGC region by the minimum requirement of a gene containing at least one condensation and one adenylation domain. NRP metallophore BGCs are technically detected by this rule as well; however, NRPSs also produce many other families of compounds, and additional manual annotation has still been required to identify NRP metallophore BGCs specifically. Accordingly, accurate prediction of BGCs encoding siderophores and other metallophores was limited to experts in natural product biosynthesis, and even experts cannot manually curate the thousands of BGCs produced by high-throughput metagenomic or comparative genomic analyses. To date, no global analysis of NRP metallophores has been performed, and thus the prevalence, combinatorics, and taxonomic distribution of different chelating groups are unknown.

Here, we describe the development and application of a high-accuracy antiSMASH-integrated method for the automated detection of NRP metallophore BGCs, using the presence of chelator biosynthesis genes within NRPS BGCs as key markers for predicting metallophore production. The new detection rule was applied to 15,562 representative bacterial genomes, allowing us to take the first census of NRP metallophore production across bacteria. At least 25% of all NRPS clusters in these representative genomes code for the production of metallophores and significant biosynthetic

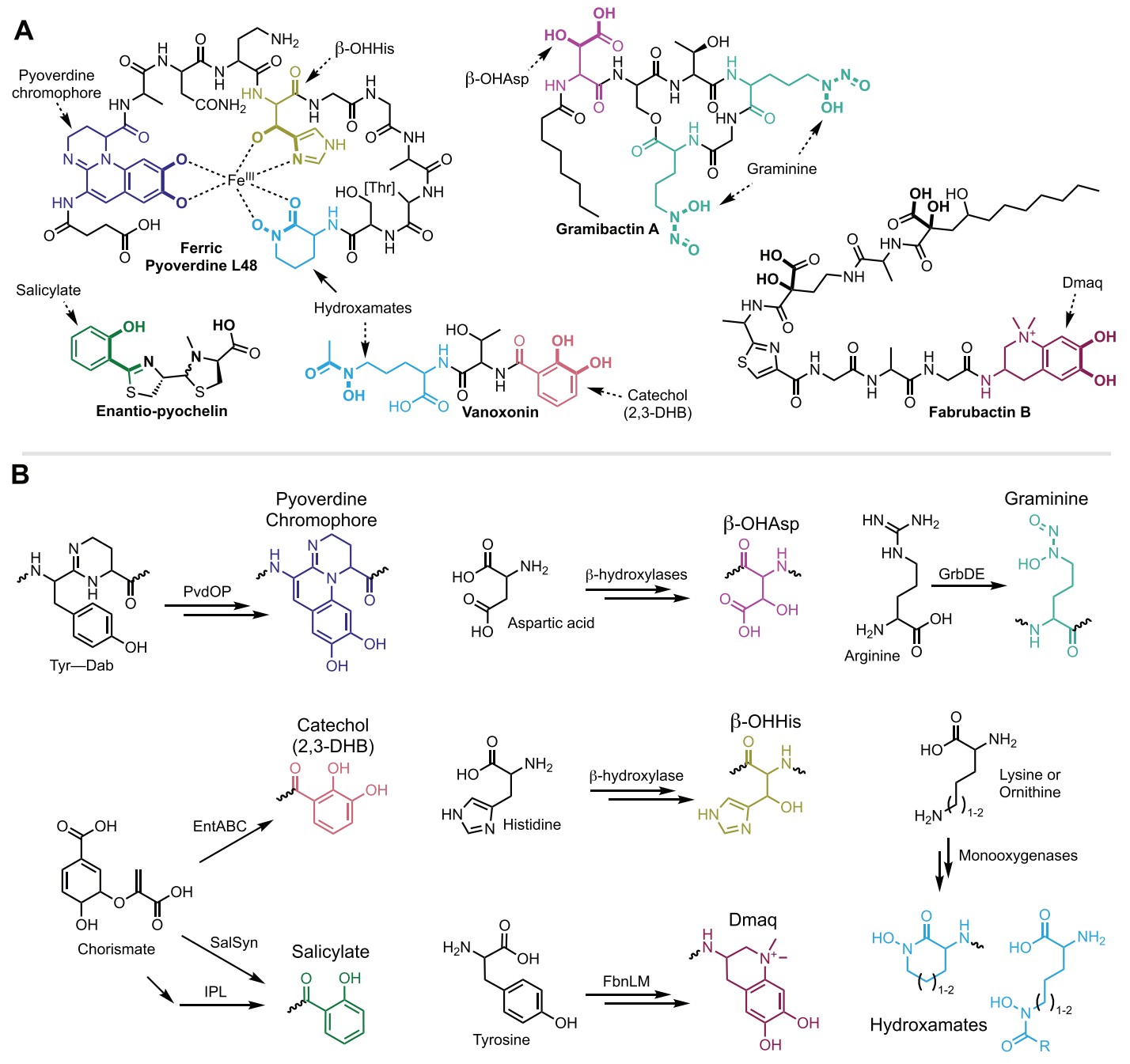

**Figure 1.** Chelating substructures found in bacterial NRP metallophores and their biosynthetic pathways. (**A**) Representative NRP metallophore structures. Nearly all known NRP metallophores contain one or more of the eight labeled chelating groups. Most chelating groups provide bidentate metal chelation, as shown for ferric pyoverdine L48. (**B**) Chelator biosynthesis pathways that form the basis for the new antiSMASH detection algorithm, as described in the text. The same chelator colors are used in each figure.

The online version of this article includes the following figure supplement(s) for figure 1:

**Figure supplement 1.** Workflow for developing an NRP-metallophore-specific profile hidden Markov model (pHMM) and significance score cutoff for an enzyme (sub-)family.

**Figure supplement 2.** A maximum-likelihood phylogeny of putative β-hydroxylases found in NRPS BGC regions.

**Figure supplement 3.** Analysis of a novel putative NRP metallophore BGC from *Sporomusa termitida* DSM 4440.

diversity remains undiscovered. We then leveraged our computational analyses to guide characterization of siderophores from multiple bacterial taxa, finding structures that matched our genome-based predictions. By mapping NRP metallophore BGCs from 59,851 genomes to the Genome Taxonomic Database (GTDB) phylogeny, we identified myxobacterial and cyanobacterial metallophores as understudied and reconstructed a possible evolutionary history of the chelating groups.

## Results

### A chelator-based strategy for detection of NRP metallophore biosynthetic gene clusters

The specialized chelating moieties found in NRP metallophores are rarely found in other natural products, and thus we sought to automate metallophore BGC prediction by searching for genes encoding their biosynthesis. An extensive review of published NRP metallophore structures revealed that nearly all contain one or more of just eight chelator substructures (*Figure 1A*). Protein domains responsible for their biosyntheses have been reported (*Figure 1B*), and thus pHMMs could be constructed to allow detection of putative chelator biosynthesis genes. Generally, draft pHMMs were built from alignments of known and predicted NRP metallophore biosynthesis genes collected from literature, and cutoffs were manually determined (*Figure 1—figure supplement 1*). The final multiple sequence alignments, pHMMs, and cutoffs are provided in the Supplemental Dataset.

A full description of each biosynthetic pathway detection strategy, including caveats and known limitations, is provided in the Methods and briefly summarized here. The profile HMMs implemented within antiSMASH are given in bold font. The biosynthetic cassette for 2,3-DHB is detected by an isochorismate synthase (**EntC**) and 2,3-dihydro-2,3- dihydroxybenzoate dehydrogenase (**EntA**) (*Raymond et al., 2003*). Two salicylate biosynthesis pathways are detected by the presence of either an isochorismate pyruvate-lyase (**IPL**) (*Serino et al., 1995*) or a bifunctional salicylate synthase (**SalSyn**) (*Pelludat et al., 2003*). We also included detection of two condensation domain subtypes specific to catecholic and phenolic metallophores: VibH-like enzymes (**VibH**) (*Keating et al., 2002*; *Reitz and Butler, 2020*) and tandem heterocyclization domains (**Cy_tandem**) (*Bloudoff et al., 2017*). Peptidic hydroxamate pathways are detected by an ornithine (Orn) or Lys N-monooxygenase (**Orn_monoox** or **Lys_monoox**, respectively) (*Olucha and Lamb, 2011*). We could not accurately detect the vicibactin hydroxylase VbsO using a pHMM (*Heemstra et al., 2009*), and so the characteristic acyl-hydroxyornithine epimerase **VbsL** is used to detect vicibactin biosynthesis (*Heemstra et al., 2009*). We previously identified three families of siderophore-specific Fe(II)/α-ketoglutarate-dependent enzymes responsible for β-OHAsp (**TBH_Asp** and **IBH_Asp**) or β-OHHis (**IBH_His**) (*Reitz et al., 2019*). Based on the recent discovery of β-OHAsp-containing cyanochelins from cyanobacteria (*Galica et al., 2021*), we have now identified two new clades that are putatively metallophore-specific and tentatively named **CyanoBH_Asp1** and **CyanoBH_Asp2**. The diazeniumdiolate-containing graminine may be detected by the presence of the cryptic necessary enzymes **GrbD** and **GrbE** (*Hermenau et al., 2019*; *Makris et al., 2022*). The quinoline chelator Dmaq is detected by **FbnL** and **FbnM**, which initiate Dmaq biosynthesis (*Vinnik et al., 2021*). The chromophore of pyoverdines is detected by the presence of a tyrosinase **PvdP** and/or an oxidoreductase **PvdO** (*Nadal-Jimenez et al., 2014*; *Ringel et al., 2018*).

Several known chelating group pathways are not currently detected. Our detection strategy is limited to clades or combinations of biosynthetic enzymes that are distinct to NRP metallophore pathways. Several chelators are synthesized by the core NRPS and/or polyketide synthase (PKS) machinery and could not be detected without also retrieving many false positives, including NRPS-derived thiazol(id)ine and oxazol(id)ine heterocycles (see pyochelin, *Figure 1A*) and PKS-derived 5-alkylsalicylate (e.g., in micocacidin *Kage et al., 2013*). We also did not include detection of a pathway currently only reported in fabrubactins that produces two α-hydroxycarboxylate chelating moieties (*Figure 1A*, bolded atoms) (*Vinnik et al., 2021*). Finally, we have not yet designed detection rules for the recently discovered chelating groups 5-aminosalicylate of pseudonochelin (*Zhang et al., 2022*) or 2-napthoate of ecteinamines (*Wu et al., 2023*); however, we expect that their biosyntheses will be amenable to detection by the method used herein (*Figure 1—figure supplement 1*). The NRP metallophore detection algorithm is publicly available in the antiSMASH web server and command line tool (https://antismash.secondarymetabolites.org/, version 7 and upwards).

**Table 1.** Summary of NRP metallophore BGC detection, comparing the chelator-based rule newly implemented in antiSMASH, the transporter-based method of Crits-Christoph et al., (*Matthijs et al., 2016*) and a combined either/or ensemble.

| | Performance metrics* | | | Number of NRP metallophore BGC regions detected in representative bacterial genomes[†] | | |
| | Precision | Recall | F1 [‡] | Complete NRPS regions n=11,704 | Partial NRPS regions n=8,403 | Total NRPS regions n=20,107 |
|---|---|---|---|---|---|---|
| AntiSMASH rule | 0.97 | 0.78 | 0.86 | 2485 (21%) [§] | 725 (8.6%) | 3210 (16%) |
| Transporter genes | 0.93 | 0.56 | 0.69 | 1723 (15%) | 376 (4.5%) | 2099 (10%) |
| Either/or ensemble | 0.92 | 0.88 | 0.90 | 2948 (25%) | 855 (10%) | 3803 (19%) |

*Detection methods were each tested on a set of 758 manually annotated NRPS BGC regions (180 true positives). Full results are given in **Supplementary file 1b**.

[†]Detection methods were applied to 15,562 NCBI RefSeq representative bacterial genomes. The full results are given in **Supplementary file 1c**. A region is 'complete' if it is not on a contig edge, as determined by antiSMASH. An additional 54 BGC regions were detected as NRP metallophores without meeting the requirements for the antiSMASH NRPS rule.

[‡]F1 score is equal to 2×(Precision ×Recall)/(Precision +Recall).

[§]Percentages indicate the fraction of NRPS regions that were predicted to encode NRP metallophores.

## Validating antiSMASH NRP metallophore detection against manually curated BGCs

In order to assess the performance of our NRP metallophore BGC detection strategy, we manually predicted metallophore production among a large set of antiSMASH-predicted BGCs. A total of 758 NRPS BGC regions from 330 genera were annotated with default antiSMASH v6.1 and inspected for known markers of metallophore production, including genes encoding transporters, iron reductases, chelator biosynthesis, and known metallophore NRPS domain motifs. We thus manually classified 176 BGC regions (23%) as metallophore BGCs (*Supplementary file 1b*). The new antiSMASH detection rule was applied to the same BGC regions, resulting in 145 putative metallophore BGC regions (F1=0.86; *Table 1* and *Supplementary file 1b*). Nine metallophore BGC regions were only detected by antiSMASH. Upon reinvestigation, four were determined to likely represent genuine metallophore BGC regions missed during manual analysis, leaving only five putative false positives in which seemingly unrelated genes matched the pHMMs (97% precision). Conversely, a total of 40 metallophore BGC regions could only be detected manually (78% recall). In the majority of false negatives, NRP metallophore BGCs were missed because chelator biosynthesis genes, on which the detection strategy is based, were not present in the cluster. In 21 cases, genes encoding catechol, salicylate, or hydroxamate biosynthesis were located elsewhere in the genome. In ten cases, chelator biosynthesis pathways were not found anywhere in the genome; these clusters may be non-functional fragments, rely on exogenous precursors (as seen in equibactin biosynthesis; *Heather et al., 2008*), or have evolved to use novel chelator biosyntheses. Two of the false negatives encoded the 5-alkylsalicylate PKS that is currently undetectable, as described above. Finally, seven manually assigned NRP metallophore BGC regions had no genes corresponding to known chelator pathways (*Supplementary file 1b*); if correctly annotated, they may represent novel structural classes. In one particularly promising case, a salicylate pathway appears to have been replaced with a partial menaquinone pathway to produce a putative 1,4-dihydroxy-2-naphthoate chelating group (*Figure 1—figure supplement 3*). A BGC from *Sporomusa termitida* DSM 4440 was annotated as a putative metallophore due to the presence of a TonB-dependent outer membrane receptor, as well as NRPS domains, methyltransferases, and oxidoreductases homologous to those involved in the biosynthesis of pyochelin and other thiazol(id)ine metallophores. A salicylate synthase gene, present in similar clusters, appears to have been replaced by a partial menaquinone pathway (*menFDEB*). We thus predicted that the salicylate moiety would be replaced by 1,4-dihydroxy-2-naphthoic acid. The Natural Product Atlas contained one family of bacterial compounds with that substructure, karamomycins, which were isolated from an unsequenced strain of *Nonomuraea endophytica* (*Figure 1—figure supplement 3*; *Winkelmann et al., 1994*). Karamomycins also contain thiazol(id)ines, as predicted for the *S. termitida* DSM 4440

BGC product. Hence, karamomycins are likely produced by a homologous BGC, and both compounds may be involved in trace metal binding and transport.

## AntiSMASH outperforms transporter-based detection, although both techniques are complementary

Crits-Christoph et al. found that the presence of transporters could be used to predict siderophore BGCs among other NRPS clusters (*Crits-Christoph et al., 2021*). Specifically, the Pfam families for TonB-dependent receptors, FecCD domains, and periplasmic binding proteins (PF00593, PF01032, and PF01497, respectively) were determined to be highly siderophore-specific, and the authors used the presence of two of the three domains to predict a 'siderophore-like' BGC region (metallophores that transport other metals were also coded as siderophores in their dataset). We used a modified version of antiSMASH to detect the three transporter families among the 758 manually annotated NRPS BGC regions (*Table 1* and *Supplementary file 1b*). In total, the transporter-based method detected 108 metallophore clusters (F1=0.69), including 8 putative false positives (93% precision), and had 80 false negatives (56% recall). One false positive was noted in the manual annotation as a likely 'cheater': while several *Bordetella* genomes encode the synthesis of a putative graminine-containing metallophore, *B. petrii* DSM 12804 has retained only the transporter genes alongside a small fragment of the NRPS. In the seven other false positives, BGC regions appeared to coincidentally contain transporter genes in their periphery, as they were not conserved in homologous NRPS clusters. In one case, the triggering genes were part of a putative vitamin B12 import and biosynthesis locus. Combining the two methods in an either/or ensemble approach slightly improved overall performance *versus* the antiSMASH rule alone, achieving 92% precision, 88% recall, and an F1 score of 0.90 (*Table 1*).

## Charting NRP metallophore biosynthesis across bacteria

The implementation of NRP metallophore BGC detection into antiSMASH allowed us to take the first bacterial census of NRP metallophore biosynthesis. The finalized detection rule was applied to 15,562 representative bacterial genomes from NCBI RefSeq (June 25, 2022). In total, 3264 NRP metallophore BGC regions were detected (*Table 1* and *Supplementary file 1c*), including 54 Type II (non- or semi-modular *Jaremko et al., 2020*) NRPS regions that would otherwise not be detected by antiSMASH, such as BGCs for acinetobactin and brucebactin (*Mihara et al., 2004*; *González Carreró et al., 2002*). NRP metallophores comprised 16% of all NRPS BGC regions in the genomes. Among complete regions (not located on a contig edge), 21% of NRPS BGC regions were classified as NRP metallophores, compared to just 8.6% of incomplete NRPS regions. This is consistent with previous reports that low-quality, fragmented genomes result in low-quality BGC annotations in antiSMASH (*Blin et al., 2017*). The transporter-based approach predicted siderophore activity for 15% of complete NRPS regions, including 463 BGC regions without detectable chelator genes; when the two methods are combined, over 25% of NRPS BGCs are predicted to produce NRP metallophores (*Table 1*). Only complete NRP metallophore BGC regions detected by antiSMASH were used for downstream analyses.

## Frequency and hybridization of NRP metallophore chelating groups

Complete NRP metallophore regions from the representative genomes were categorized by the type(s) of chelator biosynthesis genes detected within (*Figure 2*). Hydroxamates and catechols were the most common pathways, present in 44% and 36% of BGC regions, respectively. In contrast, β-OHHis, graminine, and Dmaq biosyntheses were rare in representative genomes, each present in less than 2% of detected regions. About 20% of regions contained genes for at least two pathways and putatively produce a hybrid metallophore. Only 42 BGC regions (1.7%) contained three different chelating groups: each encoded genes for the pyoverdine chromophore, a hydroxamate, and either β-OHAsp or β-OHHis. The proportion of hybrid metallophores is likely higher than estimated here. As described above, some chelating moieties could not be captured by the pHMM-based rule. Furthermore, metallophore biosynthesis may require genes from multiple BGCs. Pyoverdine genes may be located in up to five different loci (*Gross and Loper, 2009*), and all 56 regions with only the pyoverdine chromophore pathway are expected to produce hybrid siderophores. Representative characterized siderophores that contain the chelator combinations in our dataset are shown in *Figure 3*.

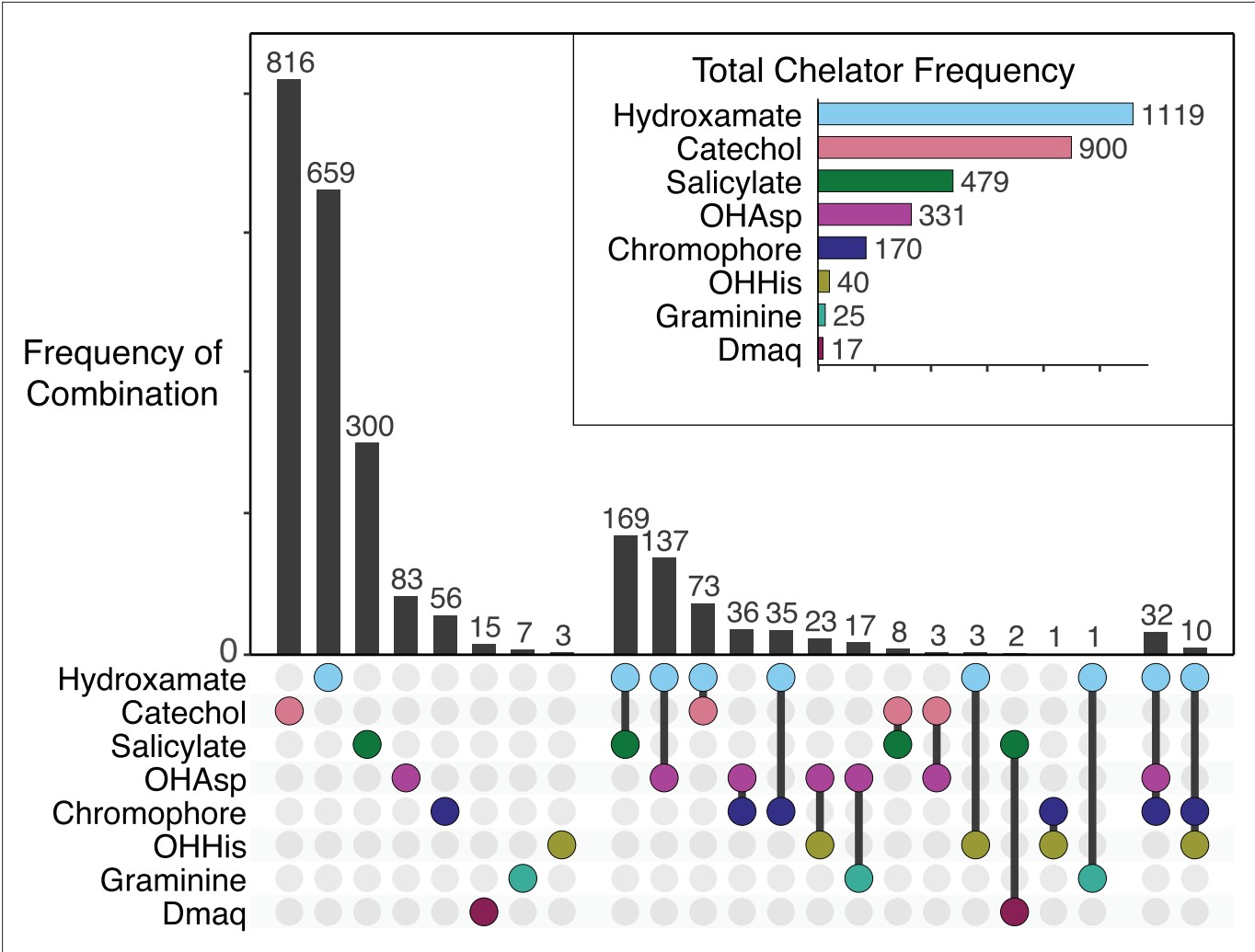

**Figure 2.** An upset plot of chelator frequency among 2,489 complete NRP metallophore BGC regions from RefSeq representative genomes. An additional 38 BGC regions were detected by metallophore-specific NRPS domains (VibH-like or tandem Cy) rather than chelator biosynthesis, and may produce catechol and/or salicylate metallophores using biosyntheses encoded elsewhere in the genome.

## The most widespread NRP metallophore families have likely been found, yet significant diversity remains unexplored

Different species of bacteria can contain highly similar metallophore BGCs. To gauge the biosynthetic diversity of the putative NRP metallophores (and thereby the structural diversity), the complete BGC regions were organized into a sequence similarity network using BiG-SCAPE, which clusters BGCs based on their shared gene content and identity. An additional 75 reported NRP metallophore BGCs were included as reference nodes (*Supplementary file 1a*), and a distance cutoff of 0.5 was chosen to allow highly similar reference BGCs to form connected components (gene cluster families; GCFs) in the network. The final network, colored and organized by chelator type, is presented in *Figure 3*. The majority of BGC regions (57%) clustered with the reference BGCs in just 45 GCFs, suggesting that many of the most widespread NRP metallophore families with known chelating groups already have characterized representatives (*Figure 3—figure supplement 1*). However, 1093 BGC regions did not cluster with any reference BGC, forming 619 separate GCFs in the network (93% of all GCFs). Some of these may encode orphan metallophores previously isolated from unsequenced strains, or be similar to known BGCs that were not included in our non-exhaustive literature search. Nevertheless, significant NRP metallophore structural diversity remains undiscovered, particularly among the 484 BGC regions distinct enough to form isolated nodes in the network.

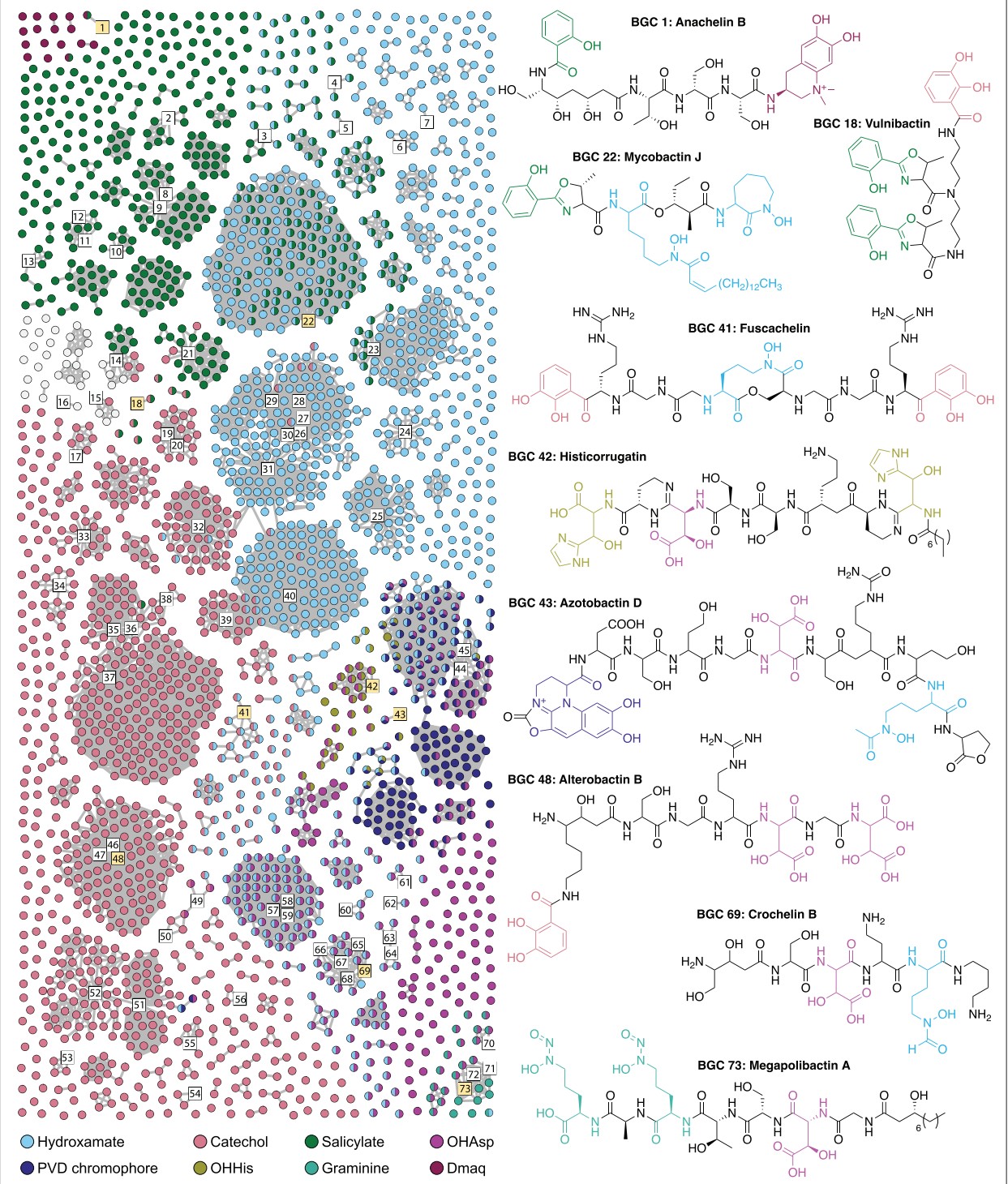

**Figure 3.** BiG-SCAPE similarity network of complete NRP metallophore BGC regions from RefSeq representative genomes. Numbered square nodes indicate published BGCs, as given in *Supplementary file 1a*. Select hybrid metallophore BGC nodes are highlighted yellow, and their corresponding structures are shown. Nodes are colored by the type(s) of chelator biosynthesis detected therein. BGC regions colored light gray contain only metallophore-specific NRPS domains (VibH-like or tandem Cy) and may produce catechol and/or salicylate metallophores using biosyntheses encoded elsewhere in the genome. The network was constructed in BiG-SCAPE v1.1.2 using 2596 BGC regions as input, including 78 reference BGCs, and a distance cutoff of 0.5.

The online version of this article includes the following figure supplement(s) for figure 3:

**Figure supplement 1.** Similarity network of complete NRP metallophore BGC regions from RefSeq representative genomes.

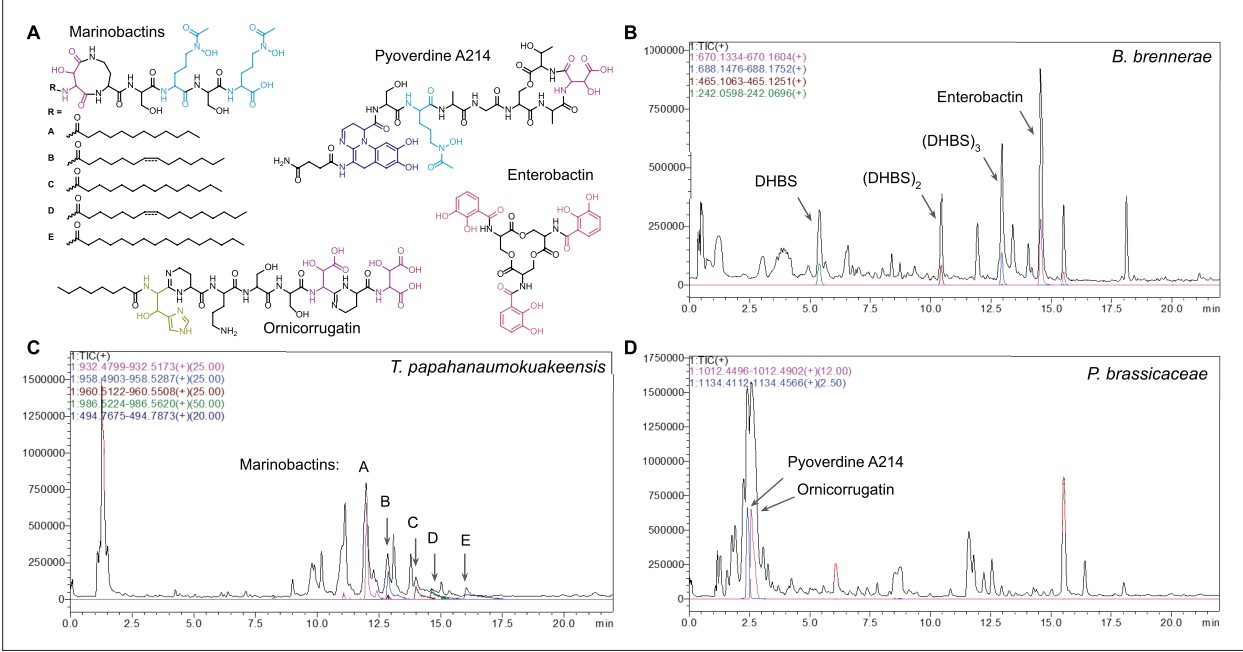

**Figure 4.** Identification of siderophores predicted from genome mining. (**A**) Chemical structures of marinobactins **A**–**E**, (**Winkelmann et al., 1994**) produced by *Terasakiispira papahanaumokuakeensis* DSM 29361; enterobactin, (**Martinez and Butler, 2007**) produced by *Buttiauxella brennerae* DSM 9396; and pyoverdine A214 (**Uría Fernández et al., 2003**) and ornicorrugatin, (**Matthijs et al., 2008**) both produced by *Pseudomonas brassicacearum* DSM 13227. The position and orientation of the fatty acid desaturation in marinobactins B and D was not determined in this work. (**B**–**D**) High pressure liquid chromatography / high-resolution mass spectrometry (HPLC-HRMS) total ion chromatograms of culture supernatant extracts, overlaid with extracted ion chromatograms for siderophore features.

The online version of this article includes the following figure supplement(s) for figure 4:

**Figure supplement 1.** Semi-preparative RP HPLC chromatogram of *B. brennerae* DSM 9396 crude extract.

**Figure supplement 2.** UPLC-ESI-MS TIC of the purified catechol compounds from *B. brennerae* DSM 9396.

**Figure supplement 3.** Mass spectra of the purified catechol compounds from *B. brennerae* DSM 9396.

**Figure supplement 4.** UPLC-ESI-MS of *T. papahanaumokuakeensis* DSM 29361.

**Figure supplement 5.** ESI-MS/MS fragmentation of marinobactin A [M+H]+, 932 *m/z*, from *T. papahanaumokuakeensis* DSM 29361.

**Figure supplement 6.** ESI-MS/MS fragmentation of marinobactin B [M+H]+, 958 *m/z*, from *T. papahanaumokuakeensis* DSM 29361.

**Figure supplement 7.** ESI-MS/MS fragmentation of marinobactin C [M+H]+, 960 *m/z*, from *T. papahanaumokuakeensis* DSM 29361.

**Figure supplement 8.** ESI-MS/MS of marinobactin D [M+H]+, 986 *m/z*, from *T. papahanaumokuakeensis* DSM 29361.

**Figure supplement 9.** UPLC-MS spectrum of ornicorrugatin siderophore from *Pseudomonas brassicacearum* DSM 13227.

**Figure supplement 10.** LC-MS tandem MS spectrum of ornicorrugatin siderophore from *Pseudomonas brassicacearum* DSM 13227.

**Figure supplement 11.** UPLC-MS spectrum of pyoverdine siderophore from *Pseudomonas brassicacearum* DSM 13227.

**Figure supplement 12.** LC-MS tandem MS spectrum of pyoverdine siderophore, pyoverdine A214, from *Pseudomonas brassicacearum* DSM 13227.

## Chemical identification of genome-predicted siderophores

To showcase how our large-scale automated genome mining methodology can be used to effectively predict functional metallophore biosynthetic pathways, we experimentally characterized the metallophores of three bacterial strains. Two strains belong to genera that have no reported metallophores: *Buttiauxella brennerae* DSM 9396 was predicted to produce enterobactin (***Figure 4A***), and *Terasakiispira papahanaumokuakeensis* DSM 29361 was predicted to produce both marinobactin(s) (***Figure 4A***) and enantio-pyochelin (***Figure 1A***). The third strain, *Pseudomonas brassicacearum* DSM 13227, was selected because its genome contains a BGC that clustered with the histicorrugatin reference BGC in the BiG-SCAPE network (***Figure 3***, node 42). We predicted that the BGC may encode the biosynthesis of ornicorrugatin (***Figure 4A***, ***Matthijs et al., 2008***) a previously reported siderophore with no known BGC. A fragmented pyoverdine BGC was also present in the strain's genome, which

was predicted to produce the known siderophore pyoverdine A214 (*Figure 4A*; *Uría Fernández et al., 2003*; *Matthijs et al., 2016*).

The crude supernatant extract from the predicted enterobactin producer *B. brennerae* was subjected to semi-preparative reverse-phase high-pressure liquid chromatography (HPLC) coupled to a UV/visible spectrophotometer. Four catecholic compounds were identified by their characteristic absorbance of UV light at 310 nm (*Figure 4—figure supplement 1*). Their relative retention times were consistent with compounds found in supernatant extracts of the known enterobactin producer *Escherichia coli*, which produces the enterobactin fragments 2,3-DHB–Ser (DHBS), $(DHBS)_2$ and linear $(DHBS)_3$, in addition to enterobactin (cyclic $DHBS_3$; *Figure 4B*; *Santichaivekin et al., 2021*). The four compounds were purified and subjected to ultra-performance liquid chromatography (UPLC) coupled to electrospray ionization mass spectrometry (ESI-MS) (*Figure 4—figure supplements 2 and 3*). The molecular ions of the purified compounds were each consistent with the enterobactin family (*m/z* 242.0647, 465.1157, 688.1614, and 670.1469, respectively).

Marinobactins A-E, differing in the identity of their fatty acid tails (*Figure 4A*), were previously isolated from *Marinobacter* spp (*Morel et al., 2024*). We searched for the compounds in the crude extract of *T. papahanaumokuakeensis* by UPLC-ESI-MS. Molecular ions consistent with all five of the marinobactins were detected (*m/z* 932.4986, 958.5159, 960.5315, 986.5422, and 988.5421, respectively, *Figure 4C*, *Figure 4—figure supplement 4*). The crude extract was then analysed by tandem ESI-MS/MS, yielding fragmentation for the peaks putatively corresponding to marinobactins A–D (the peak at *m/z* 988.5421, putatively marinobactin E, was low abundance and did not give a clear spectrum). Each had similar fragmentation patterns, with $b_4$, $b_5$, $y_1$, and $y_2$ fragments further supporting the assignments of the marinobactins (*Figure 4—figure supplements 5–8*). No peaks consistent with enantio-pyochelin (*m/z* 324.4; *Figure 1A*) could be observed.

Siderophores were likewise identified from the crude extract of *P. brassicacearum* by UPLC-ESI-MS and -MS/MS. As hypothesised, ornicorrugatin (*Figure 4A*) was indeed identified by searching the chromatogram for the expected molecular weight (*m/z* 1012.5). A candidate peak was identified with a singly, doubly, and triply charged molecular ion consistent with ornicorrugatin (*m/z* 1012.46, 506.74, and 338.17, respectively, *Figure 4D*, *Figure 4—figure supplement 9*). The tandem MS/MS spectrum closely matched that previously reported for ornicorrugatin (*Figure 4—figure supplement 10*

**Table 2.** Taxonomic distribution of 4953 NRP-metallophore BGC regions detected in 59,851 GTDB representative bacterial genomes. Phylum nomenclature is preserved from GTDB r207. An additional 413 BGC regions with 'unknown' taxonomy are not included here. Phyla not listed had zero detected regions.

| Phylum | Number of detected NRP metallophore BGC regions | Percentage of total detected NRP-met regions | Proportion of genomes with ≥1 NRP-met regions |
|---|---|---|---|
| Proteobacteria | 2439 | 49 | 2042/16536 (12%) |
| Actinomycetota | 1986 | 40 | 1561/6931 (23%) |
| Cyanobacteria | 200 | 4.0 | 176/1318 (13%) |
| Firmicutes_I | 192 | 3.9 | 191/4013 (4.8%) |
| Myxococcota | 55 | 1.1 | 52/418 (12%) |
| Firmicutes | 28 | 0.6 | 28/9026 (0.3%) |
| Chloroflexota | 18 | 0.4 | 14/1317 (1.1%) |
| Nitrospirota | 16 | 0.3 | 15/307 (4.9%) |
| Acidobacteriota | 9 | 0.2 | 9/836 (1.1%) |
| Desulfobacterota | 5 | 0.1 | 5/847 (0.6%) |
| Verrucomicrobiota | 2 | <0.1 | 2/1304 (0.2%) |
| Planctomycetota | 1 | <0.1 | 1/1034 (0.1%) |
| Bdellovibrionota | 1 | <0.1 | 1/248 (0.4%) |
| Gemmatimonadota | 1 | <0.1 | 1/345 (0.3%) |

and *Supplementary file 1d*; *Parks et al., 2022*). A chromatographic peak corresponding to pyover-dine was identified by the characteristic absorbance of the chromophore at 400 nm. A candidate compound was identified with singly and doubly charged molecular ions at *m/z* 1134.41 and 567.72, respectively (*Figure 4D*, *Figure 4—figure supplement 11*). As predicted from the BGC analysis, this mass was identical to pyoverdine A214, the previously characterized siderophore of *P. brassicacearum* subsp. *brassicacearum* NFM 421 and *Pseudomonas* sp. A214 (*Figure 4A*; *Parks et al., 2018*; *Gavrii-lidou et al., 2022*). MS/MS fragmentation yielded B-fragments identical to those previously reported (*Figure 4—figure supplement 12*; *Parks et al., 2018*).

Thus, our method was able to successfully identify the putative BGC for the orphan siderophore ornicorrugatin and also correctly predict the potential to produce known siderophores by taxa that were not yet studied for their metallophore biosynthetic capacities.

## Taxonomic distribution of NRP-metallophores

We investigated the taxonomic distribution and evolution of NRP siderophore biosynthesis within the bacterial kingdom by applying our antiSMASH detection rule to 59,851 representative bacterial genomes from the Genome Taxonomy Database (GTDB) (*Parks et al., 2022*). Among these, 4,098 genomes (6.8%) were predicted to contain at least one NRP metallophore BGC. A total of 5366 BGC regions were detected, representing 14% of all detected NRPS regions. Taxonomic distribution analysis using the GTDB phylogeny highlighted the uneven prevalence of NRP-metallophores across bacterial phyla (*Table 2*). Proteobacteria and Actinomycetota were overrepresented in the GTDB representatives, together accounting for 89% of all detected NRP metallophore regions. After correcting for the number of representative genomes in each phylum, NRP metallophore BGCs were most abundant in Actinomycetota, with 23% of genomes containing at least one detectable region. Proteobacteria, Cyanobacteria, and Myxococcota each had similar proportions of genomes with NRP metallophore BGCs; however, due to biased coverage in the GTDB database, 49% of the detected BGC regions were from Proteobacteria, compared to only 4% and 1.1% found in Cyanobacteria and Myxococcota. Thus, we expect that further sequencing efforts directed at these two phyla will yield many new NRP metallophore BGCs.

To map the distribution of NRP-metallophore producers across the bacterial kingdom, we employed Relative Evolutionary Divergence (RED) values, a framework proposed by *Parks et al., 2018* and utilized within the GTDB. Building on this, *Gavriilidou et al., 2022* defined REDgroups—phyloge-netically consistent clusters based on RED values—that provide a standardized framework analogous to genera. Unlike traditional genera, which can vary significantly in their evolutionary distances, REDgroups offer greater consistency in evolutionary relationships among their members. This framework allowed us to summarize the data as the average number of NRP-metallophore BGC regions per genome within each group, enabling effective visualization and more equitable comparative analyses of biosynthetic potential across bacterial lineages. By collapsing the GTDB tree to the REDgroup level, we annotated each group with the average number of putative NRP-metallophore clusters (*Figure 5*, Ring C). The analysis revealed that 16% of REDgroups encoded detected NRP-metallophores, and within each REDgroup, the number of NRP-metallophores was relatively consistent (standard deviation: 0.3425). This observation aligns with the findings of *Gavriilidou et al., 2022*, who demonstrated that BGC diversity is consistent at the genus level. While most REDgroups with NRP-metallophores averaged one per genome, several REDgroups, particularly within Actinomycetota, Proteobacteria, and Cyanobacteria exhibited higher averages, with some exceeding two per genome (*Figure 5*, Ring C). These results reveal lineage-specific patterns in siderophore biosynthesis and highlight the utility of REDgroups as an alternative to traditional taxonomic units.

## Evolution of gene families and phylogenetic reconciliation to uncover the evolutionary history of NRP-metallophores

To investigate the evolution and origins of NRP-metallophores, we conducted a detailed phyloge-netic analysis of each chelator group. Elucidating the evolutionary history of bacterial gene families is complicated by gene duplications, horizontal gene transfers (HGTs), and deletions that cause discordance between the bacterial species phylogeny and each chelator gene phylogeny. To reconcile the trees, we used the software package eMPRess, which infers the most likely series of duplication, HGT, and deletion events (maximum parsimony reconciliation) to reconstruct the evolutionary history

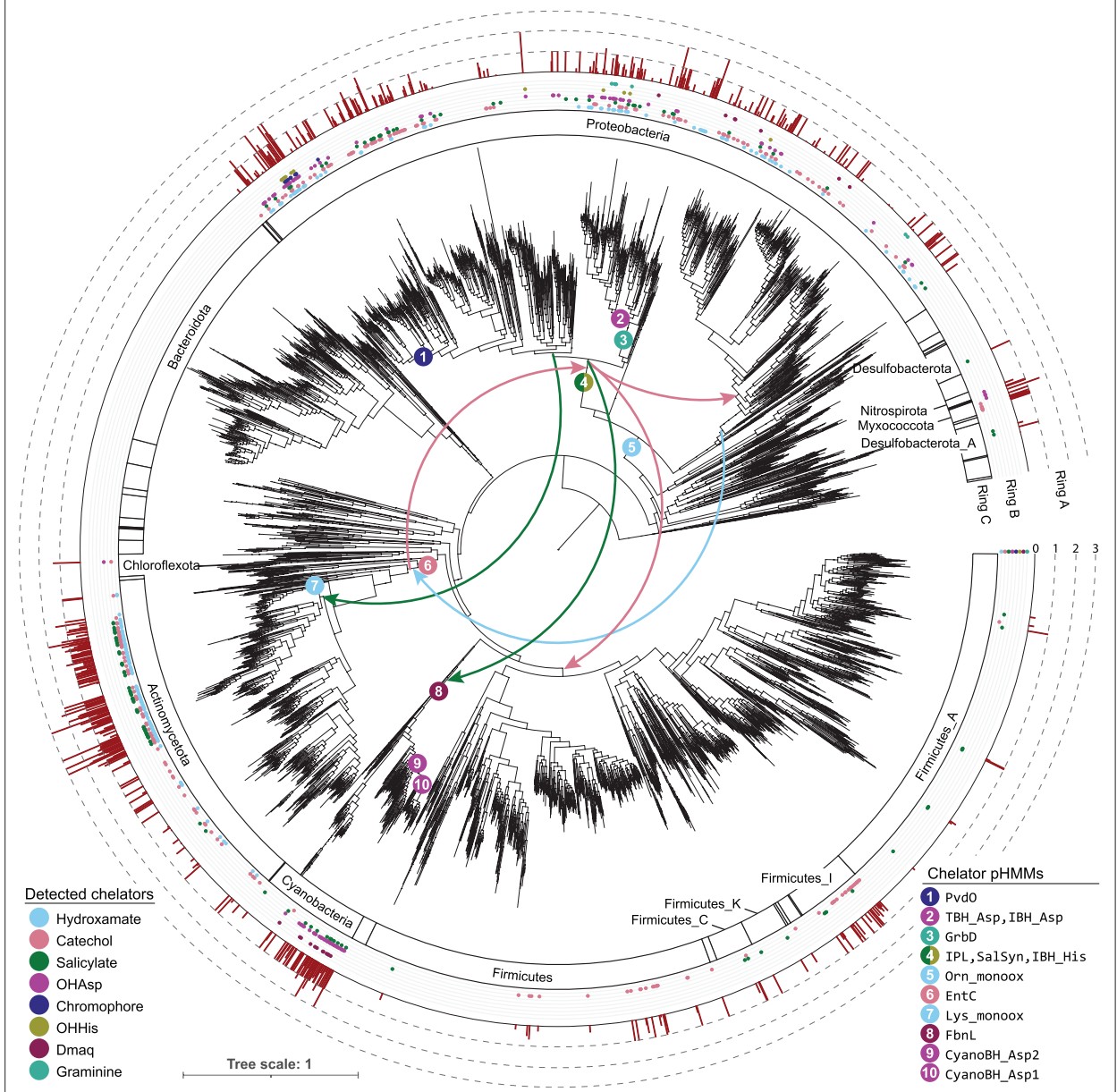

**Figure 5.** NRP metallophore biosynthesis across the bacterial kingdom. **Center:** The Genome Taxonomy Database (GTDB) phylogenetic tree (version r207), with strains collapsed to the REDgroup level (***Gavriilidou et al., 2022***). Numbered circles indicate the most parsimonious origins of chelator pathways, as determined by reconciliation with eMPRess (***Santichaivekin et al., 2021***). The bottom-right legend lists the specific hidden Markov models (pHMMs) associated with each estimated origin. Arrows indicate ancient horizontal gene transfers predicted by eMPRess. **Ring A:** Phylal divisions. Phyla with detected chelating groups are labeled using nomenclature from GTDB r207. **Ring B:** Chelator biosynthetic pathways detected in at least one member of each REDgroup. **Ring C:** Average number of detected NRP metallophore BGC regions per genome for each REDgroup. Annotations were mapped to the phylogenetic tree using iTOL v6 (***Letunic and Bork, 2024***).

of the gene family (***Santichaivekin et al., 2021***). We first extracted non-fragmented BGC regions from the GTDB representative genomes, then clustered them with BiG-SCAPE to yield 1108 representative BGCs. From these BGCs, we extracted chelator (***Morel et al., 2024***) biosynthesis genes and reconstructed gene trees, which were then compared to the GTDB species tree with eMPRess (***Santichaivekin et al., 2021***). We validated the eMPRess results by analyzing two gene families with the likelihood-based tool AleRax (***Morel et al., 2024***), which produced duplication, transfer, and loss (DTL) patterns consistent with those obtained from eMPRess.

Estimates for the origins and early HGTs of the chelating groups are presented in the center of *Figure 5*. Reconciliation indicates that the most wide-spread chelating groups—catechols, hydroxamates, and salicylates—are among the most ancient. Genes for producing 2,3-DHB may have originated in a common ancestor of Actinobacteriota (ca. 2.7 Ga, according to rough estimates from TimeTree; *Marin et al., 2017*) and were then transferred stepwise to Proteobacteria and to Firmicutes. Salicylate biosynthesis genes have an estimated origin in a common ancestor of Gammaproteobacteria (ca. 1.9 Ga; *Marin et al., 2017*), with early HGT to Cyanobacteria and Actinobacteriota. Hydroxamate NRP metallophores appear to have originated in the common ancestor of Alpha- and Gammaproteobacteria (ca. 2.3 Ga; *Marin et al., 2017*) and were transferred into Actinobacteria, while Lys-based hydroxamates evolved in Actinobacteriota. The other chelator groups display a more phylum-specific distribution, with HGT predominantly occurring within the same phylum (see Supplemental Dataset, empress_reconciliations). Dmaq is predicted to be among the oldest chelating groups and may have been produced by the common ancestor of Cyanobacteria (ca. 2.7 Ga; *Marin et al., 2017*), while the pyoverdine chromophore, exclusively observed within the order Pseudomonadales, likely represents one of the most recent siderophore biosynthetic pathways (ca. 1.2 Ga; *Marin et al., 2017*).

## Discussion

Trace metal starvation shapes interactions within microbial communities and between bacteria and the host; therefore, natural and synthetic microbiomes cannot be understood without knowing the metallophore biosynthetic potential of the community. High-throughput biotechnological applications will benefit from in silico metallophore prediction due to the prohibitively high cost of isolation and characterization. To date, distinguishing peptidic metallophore BGCs from other NRPS BGCs has been largely limited to manual expert analyses, leading to blind spots in our understanding of microbes and their communities. We have now automated bacterial NRP metallophore prediction by extending the secondary metabolite prediction platform antiSMASH to detect genes involved in the biosynthesis of metal chelating moieties, enabling the first global analysis of bacterial metallophore biosynthetic diversity.

The presence of genes encoding catechols, hydroxamates, and other chelating groups is one of the most frequently used markers of a metallophore BGC (*Reitz and Medema, 2022*). We have formalized and automated the identification of eight chelator pathways, allowing us to detect 78% of NRP metallophore BGCs with a 3% false-positive rate against a manually annotated set of NRPS clusters. Biosynthetic genes are detected with custom pHMMs and significance score cutoffs calibrated for accurate metallophore discovery, diminishing the ambiguity of interpreting gene annotations, protein families, and BLAST results. We acknowledge that human biases may have influenced which clusters were coded as putative metallophores during both algorithm development and testing; however, expert manual curation remains the most reliable method for NRP metallophore BGC detection. Unfortunately, 22% of manually identifiable metallophore BGCs could not be automatically distinguished from other NRPS clusters, as the algorithm developed (for the purpose of being easily integrated into antiSMASH) relies on the presence of one or more known chelator biosynthesis genes colocalized with the NRPS genes. Detection rates were also lower for fragmented genomes, a limitation (inherent to antiSMASH itself; *Blin et al., 2017*) that hinders the identification of metallophore biosynthesis in metagenomes. As long-read sequencing of metagenomes becomes more common, we expect that detection will improve.

Recently, Crits-Christoph et al. demonstrated the use of transporter families to predict that a BGC encodes siderophore (or metallophore) biosynthesis (*Crits-Christoph et al., 2021*). Among our test dataset, the biosynthesis-based antiSMASH rule outperformed the transporter-based approach (F1=0.86 versus F1=0.69). However, some putative metallophore BGCs were only found using the transporter-based approach, and a combined either/or ensemble approach slightly outperformed the antiSMASH rule alone (F1=0.90). Biosynthetic- and transporter-based techniques are thus complementary, and future work could incorporate transporter genes into antiSMASH metallophore prediction. We note that the reported transporter-based approach uses just three pHMMs from Pfam, while our biosynthetic detection requires many custom pHMMs. An extended set of metallophore-specific transporter pHMMs designed according to the same principles as those followed for the biosynthesis-related pHMMs could significantly improve detection by reducing false positives and capturing other

families of transporters. The NRP metallophore BGCs discovered in this study could serve as a dataset for developing a more comprehensive model for metallophore transporter detection.

The diverse enzyme families responsible for the biosynthesis of NRP metallophore chelating groups (*Figure 1B*) evince that metal chelation has evolved multiple times, and we expect that more NRPS chelator substructures remain undiscovered. In fact, during manuscript preparation, the novel chelator 5-aminosalicylate was reported in the structure of the *Pseudonocardia* NRP siderophore pseudonochelin (*Zhang et al., 2022*), and we found several unexplored clades of Fe(II)/α-ketoglutarate-dependent amino acid β-hydroxylases that are likely involved in metallophore biosyntheses (*Figure 1—figure supplement 2*). Additionally, we have likely identified a new biosynthetic pathway in the genome of *Sporomusa termitida* DSM 4440, which encodes a partial menaquinone pathway in place of a salicylate synthase to putatively produce a novel karamomycin-like metallophore (*Figure 1—figure supplement 3*; *Shaaban et al., 2019*). The modular nature of the pHMM-based detection rule will allow for new chelating groups to be added as their biosyntheses are experimentally characterized.

Metallophore BGC regions from representative genomes were compared to reference BGCs and de-replicated into gene cluster families (GCFs) with BiG-SCAPE (*Figure 3*). We found 1093 metallophore BGC regions that were dissimilar from any reference BGC, and almost 500 distinct BGC regions were found in only a single strain. Although significant biosynthetic diversity remains undiscovered, cluster de-replication will become increasingly important to avoid re-isolating known compounds. We also assessed the taxonomic distribution of NRP-metallophore BGC regions by mapping their presence onto a GTDB REDgroup phylogeny. We found that Cyanobacteria and Myxococcota were underrepresented in our analyses due to a relatively low number of published genomes. Considering that only a handful of NRP metallophores have been isolated from these two phyla, we suggest that Cyanobacteria and Myxococcota deserve coordinated efforts of genomic sequencing and experimental work to further characterize their metallophore diversity.

Finally, we used our dataset of detected BGCs and paired taxonomic data from GTDB to investigate the complex evolutionary history of chelating group biosynthesis by reconstructing the most likely origin and major HGT events for each pathway with eMPRess (*Figure 5*; *Santichaivekin et al., 2021*). Catechols, hydroxamates, and salicylates were among the most widespread and ancient chelators with evidence of HGT between phyla. This widespread distribution suggests significant ecological relevance for these chelators in diverse bacterial lineages. Intriguingly, our timeline estimates place the origin of 2,3-DHB and Dmaq prior to the Great Oxygenation Event (~2.4–2.1 Ga), during an era of abundant, soluble ferrous iron. This result leads credence to the hypothesis that chelating molecules first evolved as metal detoxification mechanisms and were repurposed when oxidized iron became scarce (*Kramer et al., 2020*). Tracing ancient evolutionary events, particularly those involving multiple gene gains and losses, remains challenging due to the exponential increase in complexity as the number of possible events grows. More detailed examinations dedicated to each individual chelating group may yield deeper insights into the complex evolutionary history of these pathways. For example, the origin of hydroxamates must consider the homologous enzymes in NRPS-independent siderophore pathways, and we cannot yet state if metallophore-specific β-OHAsp biosynthesis is polyphyletic due to repeated incorporation into metallophores or a single incorporation followed by repeated transfer into non-chelating roles. Nevertheless, this study represents, to our knowledge, the first attempt at a large-scale phylogenetic analysis into the origin of chelating groups in bacteria.

By integrating chelator detection into antiSMASH, we have taken a major step towards accurate, automated NRP metallophore BGC detection. The new strategy affords a clear practical improvement over manual curation, and has already allowed for the high-throughput identification of thousands of likely NRP metallophore BGC regions, both in this study and in several other recently published analyses that have been enabled by early availability of our methodology (*Mohite et al., 2024*; *Jørgensen et al., 2024*). A future antiSMASH module might predict metallophore activity more accurately with a machine learning algorithm that considers multiple forms of genomic evidence, including the presence of transporter genes, NRPS domain architecture and sequence, metal-responsive regulatory elements, and other markers of metallophore biosynthesis that are still limited to manual inspection (*Reitz and Medema, 2022*). In particular, regulatory elements will likely be required to accurately distinguish siderophores, zincophores, and other classes of metallophores. We also envision a later version that predicts specific structural features of the metallophore by identifying accessory biosynthetic genes. Implementation of NRP metallophore BGC detection into antiSMASH will enable

scientists of diverse expertises to identify and quantify NRP metallophore biosynthetic pathways in their bacterial genomes of interest and promote large-scale investigations into the chemistry and biology of metallophores.

## Methods

For all software, default parameters were used unless otherwise specified. All python, R, and bash scripts used in this paper, as well as underlying data, is available in the Supplemental Dataset, published to Zenodo: 10.5281/zenodo.17806971. All strains were obtained from the Leibniz Institute DSMZ. Unless otherwise stated, all water is doubly deionized (dd $H_2O$), 18 M$\Omega$.

### Profile hidden Markov model construction

Profile hidden Markov models (pHMMs) were built from biosynthetic genes in known metallophore pathways, supplemented with putative BGC genes where required (*Figure 1—figure supplement 1* and Supplemental Dataset, 1_development/). Amino acid sequences were aligned with MUSCLE (v3.8) (*Edgar, 2004*) and pHMMs were constructed using hmmbuild (HMMER3) (*Eddy, 2011*). pHMMs were tested against the MIBiG database (v2.0) (*Kautsar et al., 2020*) and an additional 37 NRP sidero-phore BGCs from literature (*Supplementary file 1a*) using hmmsearch (HMMER3). Rough bitscore significance cutoffs were determined for each pHMM. More precise cutoffs were assigned by testing against 28,688 NRPS BGC regions from the antiSMASH database (v3) (*Blin et al., 2021a*). BGC regions containing genes near the rough cutoff were manually inspected to determine if these were likely metallophore BGCs based on the presence of genes encoding membrane transport, metal acquisition (ex: ferric reductases), and multiple chelating groups. If no clear bitscore cutoff could be discerned, representative low-scoring putative true hits were added to the pHMM seed alignment. This process was repeated until a precise bitscore cutoff could be determined.

### Detection of catechols and phenols

Like the aromatic amino acids, 2,3-dihydroxybenzoic acid (2,3-DHB) and salicylic acid are derived from chorismate (*Figure 1B*; *Raymond et al., 2003*; *Serino et al., 1995*; *Pelludat et al., 2003*). The three-step synthesis of 2,3-DHB is catalyzed by an isochorismate synthase (**EntC**), an isochorismatase (EntB), and a 2,3-dihydro-2,3- dihydroxybenzoate dehydrogenase (**EntA**) (*Raymond et al., 2003*). The sole presence of **EntA** matches in a cluster was sufficient to detect 2,3-DHB-containing metallophore BGCs; however, only **EntC** could be used to eliminate the nearly-identical 3-hydroxyanthranilic acid pathway (*Pavlikova et al., 2018*), and thus both **EntA** and **EntC** were required to accurately detect 2,3-DHB production. Salicylate biosynthesis was detected by the presence of either an isochorismate pyruvate-lyase (**IPL**) (*Serino et al., 1995*) or a bifunctional salicylate synthase (**SalSyn**) (*Pelludat et al., 2003*). Two condensation domain subtypes specific to catecholic and phenolic metallophores were also included as independent detection rules: VibH-like enzymes (**VibH**), which condense 2,3-DHB to diamines and polyamines, (*Keating et al., 2002*; *Reitz and Butler, 2020*) and tandem heterocycliza-tion domains (**Cy_tandem**) are sometimes responsible for Ser, Thr, or Cys cyclization (*Bloudoff et al., 2017*).

### Detection of hydroxamic acids

Hydroxamates are all produced by the hydroxylation and acylation of a primary amine. In peptidic metallophores, the amine is usually the sidechain of ornithine (Orn) or Lys, which is first hydroxylated by a flavin-dependent monooxygenase (**Orn_monoox** or **Lys_monoox**, respectively). Vicibactin is an unusual hydroxamate siderophore (*Heemstra et al., 2009*) that could not be accurately captured by either **Orn_monoox** or **Lys_monoox**. The Orn hydroxylase VbsO is more similar to those found in NIS pathways, and a VbsO pHMM was non-specific. Instead, the acyl-hydroxyornithine epimerase (*Heemstra et al., 2009*) **VbsL** is used to detect vicibactin. In two non-metallophore pathways (*Du et al., 2017*; *Matsuda et al., 2022*), hydroxyornithine or hydroxylysine are the substrates for N-N bond-forming enzymes. No clear bitscore separation could be achieved to eliminate these false posi-tives, so two negative constraints were added: the presence of **KtzT** associated with biosynthesis of piperazates (*Du et al., 2017*), and **MetRS-like**, associated with several hydrazines (*Matsuda et al., 2022*). Unfortunately, false positives may still arise if the ornithine monooxygenase and the negative

constraint genes are distantly located within the same BGC region (>30 kbp), such as in the himastatin locus (MIBiG BGC0001117).

## Detection of graminine, Dmaq, and the pyoverdine chromophore

Although the biosynthesis of the chelating amino acid graminine has not been fully elucidated, gene knockouts and stable isotope studies have revealed two enzymes, **GrbD** and **GrbE**, responsible for diazeniumdiolate formation from arginine (*Hermenau et al., 2019*; *Vinnik et al., 2021*) The quinoline chelator Dmaq was first identified in anachelins (*Beiderbeck et al., 2000*) and the biosynthesis was recently established for fabrubactins (*Vinnik et al., 2021*). Synthesis is initialized by **FbnL** and **FbnM**, which form a two-protein heme peroxidase that oxidizes L-tyrosine to L-DOPA (*Vinnik et al., 2021*). GrbDE and FbnLM were previously used as handles for genome mining (*Hermenau et al., 2019*; *Vinnik et al., 2021*), and our pHMMs gave similar results. We did not include detection of a pathway currently only reported in fabrubactins that produces two α-hydroxycarboxylate chelating moieties (*Figure 1A*, bolded atoms) (*Vinnik et al., 2021*). Both substructures are proposed to be synthesized by the flavin-dependent monooxygenase FbnE. A profile HMM was built for FbnE (Supplemental Dataset) but was not included in antiSMASH, as all BGCs with **FbnE** hits were either captured independently by **FbnL** and **FbnM** (Dmaq biosynthesis), or appeared to be false positives based on a lack of other metallophore-related genes. The diverse pyoverdine family of siderophores is defined by a fluorescent chelating chromophore (*Figure 1*). Chromophore maturation is driven by the tyrosinase **PvdP** and the oxidoreductase **PvdO** (*Nadal-Jimenez et al., 2014*; *Ringel et al., 2018*). The two genes are sometimes in separate loci, and including both as independent pHMMs increased the number of pyoverdine BGCs detected. These pHMMs also capture the biosynthesis pathway for azotobactin, which contains a slightly different chromophore (*Figure 3*; *Demange et al., 1988*).

## Detection of β-hydroxyamino acids and phylogenetic analysis of Asp and His β-hydroxylases

Our previous analysis of siderophores containing β-hydroxy-aspartate (β-OHAsp) and β-hydroxy-histidine (β-OHHis) delineated three families of siderophore-specific Fe(II)/α-ketoglutarate-dependent enzymes responsible for β-hydroxylation of Asp (**TBH_Asp** and **IBH_Asp**) or His (**IBH_His**) (*Reitz et al., 2019*). More recently, β-OHAsp-containing siderophores named cyanochelins were isolated from a diverse group of cyanobacteria (*Galica et al., 2021*). The BGCs contain two additional classes of β-hydroxylases, which we have tentatively named **CyanoBH_Asp1** and **CyanoBH_Asp2**. Because of the polyphyletic nature of these hydroxylases, well-performing pHMMs could not be made using the standard workflow used for other pathways (*Figure 1—figure supplement 1*); even after many iterations, non-metallophore NRPs were captured as false positives, such as the β-OHAsp-containing lipopeptide turnercyclamycin (*Miller et al., 2021*).

An expanded phylogenetic analysis was performed to serve as a guide for pHMM construction (Supplemental Dataset, 1_development/hydroxylase_tree/). NRPS BGC regions from the antiSMASH database (v3) were scanned for matches to previously reported β-hydroxylase pHMMs (*Reitz et al., 2019*) and Pfam pHMMs for siderophore-related transporters (PF00593, PF01032, and PF01497; *Crits-Christoph et al., 2021*; *El-Gebali et al., 2019*) using a modified version of antiSMASH v6.0. β-Hydroxylase genes meeting a relaxed bitscore cutoff of 300 (1070 genes total) were dereplicated with CD-HIT web server (*Huang et al., 2010*) and a sequence identity cutoff of 70%, giving 425 representative amino acid sequences. A multiple sequence alignment was created using hmmalign (HMMER3) and the TauD Pfam (PF02668) (*El-Gebali et al., 2019*) and a maximum-likelihood phylogenetic tree was reconstructed with IQ-TREE (multicore v2.2.0-beta) (*Nguyen et al., 2015*) using the WAG +F + I+G4 evolutionary model. The presence of nearby transporters was mapped onto the phylogenetic tree to identify clades or paraphyletic groups putatively involved in siderophore biosynthesis. Sequences in groups corresponding to previously reported TBH_Asp, IBH_Asp, and IBH_His subtypes and the novel putative CyanoBH_Asp1 and CyanoBH_Asp2 subtypes were extracted, and pHMMs were constructed and tested as described above. Several unreported clades of amino acid β-hydroxylases were identified that may also be involved in metallophore biosynthesis based on their co-occurrence with transporter genes (*Figure 1—figure supplement 2*). However, these putative β-hydroxylases were not included in the current NRP metallophore rules due to a lack of experimental evidence for metallophore production. A negative constraint (in the form of a competing pHMM for

**SBH_Asp**) was used to exclude the syringomycin family of BGCs, which contain a clade of β-hydroxy-lases that sit within the IBH_Asp clade (*Reitz et al., 2019*).

## Incorporation into antiSMASH

The pHMMs and cutoffs were added to antiSMASH as a single detection rule called 'NRP-metallophore' with the following logic:

```
VibH_like or Cy_tandem or
(cds(Condensation and AMP-binding) and (
    (IBH_Asp and not SBH_Asp) or IBH_His or TBH_Asp or
    CyanoBH_Asp1 or CyanoBH_Asp2 or
    IPL or SalSyn or (EntA and EntC) or
    (GrbD and GrbE) or (FbnL and FbnM) or PvdO or PvdP or
    (Orn_monoox and not (KtzT or MetRS-like))
    Lys_monoox or VbsL))
```

## Manual validation

A subset of RefSeq representative bacterial genomes was generated by randomly selecting one genome for each of the 330 genera determined by GTDB (Supplemental Dataset, 3_manual_testing/). All NRPS BGC regions in the genomes were annotated with antiSMASH v6.1, yielding 758 BGC regions in the final testing dataset (*Supplementary file 1b*). The antiSMASH output for each BGC was manually inspected for evidence of NRP metallophore production, including genes encoding transporters, iron reductases, chelator biosynthesis, and known metallophore NRPS domain motifs. The same 758 BGC regions were classified as NRP metallophores using the chelator-based strategy described above. We compared the performance against the transporter-based strategy of *Crits-Christoph et al., 2021*. Genes matching Pfam pHMMs for siderophore-related transporters (PF00593, PF01032, and PF01497) (*Crits-Christoph et al., 2021*; *El-Gebali et al., 2019*) were identified using a custom version of antiSMASH v6.1, in which the three pHMMs were added to the list of domains that are identified and labeled as 'biosynthetic-additional'. BGC regions were classified as metallophores if two of the three transporter families were present (*Crits-Christoph et al., 2021*). Each putative false positive was re-investigated before performance statistics were calculated, resulting in the reannotation of four BGCs.

## BIG-SCAPE clustering

The 3264 NRP metallophore BGC regions from RefSeq representative genomes (Supplemental Dataset, 2_refseq_reps_results/metallophores_Jun25.tar.gz) were filtered to remove BGC regions on contig boundaries. The resulting 2523 BGC regions, as well as 78 previously reported BGCs (*Supplementary file 1b*) were clustered using BiG-SCAPE v1.1.2 with the following settings: "`--no_classify --mix --cutoffs 0.3 0.4 0.5 --clans-off`". The network (Supplemental Dataset, 6_bigscape/mix_c0.50.network) was imported to Cytoscape for figure preparation.

## Bacterial culturing and siderophore isolations

*Terasakiispira papahanaumokuakeensis* DSM 29361 and *Buttiauxella brennerae* DSM 9396 were cultivated in a 4 L Erlenmeyer flask acid washed with 4 M HCl. *T. papahanaumokuakeensis* was grown in 2 L of the following medium: glycerol phosphate disodium 4 g/L, NaCl 25 g/L, KCl 0.67 g/L, CaCl$_2$ * 2 H$_2$O 1.36 g/L, NH$_4$Cl 4 g/L, L sodium succinate hexahydrate 5 g/L and 25 mM MOPS pH 7.5. For *B. brennerae*, the medium was 2 L: K$_2$HPO$_4$ 7 g/L, KH$_2$PO$_4$ 2 g/L, NaCl 0.6 g/L, MgSO$_4$ * 7 H$_2$O 1 g/L, NH$_4$SO$_4$ 1 g/L sodium succinate hexahydrate 5 g/L. After autoclaving 20 mL of filter sterilized D-glucose 50% (w/v) was added, completing the medium. The cultures were incubated at room temperature on an orbital shaker, 180 RPM. After 41 h and 48 h, the cultures of *T. papahanaumokuakeensis* and *B. brennerae*, respectively, were centrifuged at 6,000xg for 10 min at 4°C. The supernatant was decanted and approximately 100 g of XAD-4 resin added. The solution was agitated gently at room temperature for approximately 4 h. Organic compounds were eluted from the resin with 100% methanol, which after rotary evaporation yielded the crude extract. Four catechol compounds produced by

*B. brennerae* were isolated using reverse phase (RP) HPLC. The HPLC system was comprised of a YMC AQ12S05-2520WT column connected to two Waters 515 HPLC pumps and a waters 2487 dual wavelength absorbance detector. The mobile phase consisted of dd $H_2O$ and methanol, both amended with 0.05% trifluoroacetic acid (w/v). To separate the compounds a method of 10–100% methanol, 3% per min was utilized. The crude extract of *T. papahanaumokuakeensis* and the catechol compounds of *B. brennerae* were analyzed by ESI-MS and MS/MS on a Waters XevoG2-XS QTof instrument in positive mode electrospray ionization coupled to an ACQUITY UPLC H-Class system with a Waters BEH C18 column.

*Pseudomonas brassicacearum* DSM 13227 was cultured in a modified M9 minimal medium composed of 3.0 g/L disodium hydrogen phosphate heptahydrate, 1.5 g/L potassium dihydrogen phosphate, 2.5 g/L sodium chloride, 0.50 g/L ammonium chloride, 2.5 g/L disodium succinate, and 5.0 g sodium pyruvate in ultrapure (e.g., 18 mW) water, amended after autoclave sterilization with 20 mL/L 50% w/v Steri-filtered glucose solution, 1 mL/L 1 M magnesium chloride, and 0.05 mL/L 1 M calcium chloride. Culturing glassware was rinsed with 4 M HCl before use to remove adsorbed iron(III). Seed cultures were inoculated in Luria-Bertani (LB) broth with single colonies of *P. brassicacearum* DSM 13227 grown on LB agar and grown for at least 24 h at 30°C. Cultures were monitored by $OD_{600}$, and the culture supernatant was harvested at late log/early stationary phase ($OD_{600}{\sim}1.2$ au) with a positive Fe(III)-CAS response. Culture supernatants were obtained by centrifugation at 6500 rpm for 20 min at 4°C. To extract the siderophores, the culture supernatant was decanted and shaken with 100 g of Amberlite XAD-4 resin. The XAD-4 resin was prepared by washing with methanol and then equilibrating with ultrapure water. The resin and supernatant were orbitally shaken to equilibrate for 2 h at 150 rpm. The resin was filtered from the supernatant and washed with 0.5 L of ultrapure water. The adsorbed organics were eluted with 80% aqueous methanol. The eluent was concentrated under vacuum and stored at 4°C. Ornicorrugatin and the pyoverdine siderophore were identified by UPLC-MS and LC-MS/MS analysis of the concentrated eluent.

## Phylogenetic mapping

Genome mining was performed on 62,291 GTDB representative genomes (59,851 after filtering; version r207) (*Parks et al., 2022*) using AntiSMASH v7.0beta, (*Blin et al., 2023*; *Parks et al., 2022*) with the inclusion of the NRP metallophore detection module. The outputs were analysed to identify predicted NRP-metallophore producers and categorized into distinct chelator groups based on predefined detection criteria. A total of 5366 NRP-metallophores were identified, representing approximately 14% of all detected NRPS regions. To map the distribution patterns of these producers, the results were integrated with the GTDB tree. Due to the size of the tree, visualization tools such as iTOL (*Letunic and Bork, 2024*) were impractical, prompting dereplication to a higher taxonomic rank. The GTDB tree was collapsed to the REDgroup level—a phylogenetically defined rank analogous to genera—allowing normalization to reflect the average number of NRP-metallophore biosynthetic gene clusters (BGCs) per genome within each REDgroup (*Gavriilidou et al., 2022*).

To uncover the evolutionary history of siderophore biosynthesis, phylogenetic analyses and reconciliation were performed. Gene sequences for each chelator group were extracted from 4060 complete BGCs, filtered to exclude BGC regions on contig boundaries, and clustered into Gene Cluster Families (GCFs) using BiG-SCAPE (*Navarro-Muñoz et al., 2020*) with a 0.5 cutoff. From each GCF, one representative BGC was selected, resulting in a dataset of 1108 clusters. Multiple sequence alignments (MSAs) were conducted using MAFFT v7, (*Katoh and Standley, 2013*) and phylogenetic trees were constructed using FastTree 2 with the WAG model (*Price et al., 2010*). Evolutionary events, including gene duplication, loss, and horizontal gene transfer, were identified using phylogenetic reconciliation in eMPRess (*Santichaivekin et al., 2021*) by comparing gene trees to species trees. To validate the robustness of the inferred evolutionary scenarios, SalSyn and CyanoBH_Asp1 were also reconciled using the likelihood-based tool AleRax (*Morel et al., 2024*). Reconciliation results were annotated using iTOL v6 (*Letunic and Bork, 2024*) for visualization, manually mapping key evolutionary events onto the GTDB tree. Individual gene tree reconciliations are available in the Supplementary Dataset.

## Acknowledgements

This project has received funding from the European Research Council under the European Union's Horizon 2020 research and innovation program (Starting Grant 948770-DECIPHER; ZR and MM), as

well as from the US National Science Foundation (CHE-2108596; AB). BT and NZ were supported by H2020-FNR-11-2020: SECRETED—Grant agreement: 101000794. NZ was supported by the German Center for Infection Research TTU09.716. We thank Allegra Aron for providing useful feedback on the manuscript.

## Additional information

### Competing interests

Marnix H Medema: M.H.M. is a member of the Scientific Advisory Boards of Hexagon Bio and Hothouse Therapeutics. The other authors declare that no competing interests exist.

### Funding

| Funder | Grant reference number | Author |
|---|---|---|
| European Research Council | 948770-DECIPHER | Marnix H Medema |
| National Science Foundation | CHE-2108596 | Alison Butler |
| HORIZON EUROPE Framework Programme | H2020-FNR-11-2020 | Nadine Ziemert |
| Deutsches Zentrum für Infektionsforschung | TTU09.716 | Nadine Ziemert |

The funders had no role in study design, data collection and interpretation, or the decision to submit the work for publication.

### Author contributions

Zachary L Reitz, Conceptualization, Data curation, Software, Formal analysis, Validation, Investigation, Visualization, Methodology, Writing – original draft; Bita Pourmohsenin, Conceptualization, Data curation, Formal analysis, Investigation, Visualization, Methodology, Writing – original draft; Melanie Susman, Formal analysis, Investigation, Methodology, Writing – review and editing; Emil Thomsen, Daniel Roth, Formal analysis, Investigation, Writing – review and editing; Alison Butler, Supervision, Funding acquisition, Writing – review and editing; Nadine Ziemert, Marnix H Medema, Conceptualization, Supervision, Funding acquisition, Project administration, Writing – review and editing

### Author ORCIDs

Zachary L Reitz ⓘ https://orcid.org/0000-0003-1964-8221
Alison Butler ⓘ https://orcid.org/0000-0002-3525-7864
Nadine Ziemert ⓘ https://orcid.org/0000-0002-7264-1857
Marnix H Medema ⓘ https://orcid.org/0000-0002-2191-2821

Reviewer #1 (Public review): https://doi.org/10.7554/eLife.109154.3.sa1
Reviewer #2 (Public review): https://doi.org/10.7554/eLife.109154.3.sa2
Author response https://doi.org/10.7554/eLife.109154.3.sa3

## Additional files

### Supplementary files

Supplementary file 1. Tabular data for reference BGCs, rule validation, RefSeq genome results, and ornicorrugatin MS/MS. An Excel (xlsx) file containing four sheets: (a) Reference BGCs used in this work. Numbering matches *Figure 3*. BGCs marked with an asterisk are non-metallophore BGCs that emerged as false positives during rule development. (b) Data for the rule validation. The sheet contains BGC metadata, notes used for manual classification, and individual results from detection rule strategies, as well as summary confusion matrices and performance metrics. (c) Results for applying detection rules to BGCs from NCBI RefSeq representative bacterial genomes, including

bitscore values for significant matches to profile HMMs. (d) Selected MS/MS fragment ions observed for ornicorrugatin produced by Pseudomonas brassicacearum DSM 13227. All given fragment ions agree with previously reported masses for ornicorrugatin (*Matthijs et al., 2008*).

MDAR checklist

## Data availability

All python, R, and bash scripts used in this paper, as well as underlying data, is available in the Supplemental Dataset, published to Zenodo: 10.5281/zenodo.18866949. The enterobactin, marinobactin, and ornicorrugatin BGCs have been submitted to the MIBiG repository (*Blin et al., 2021a*) with accession numbers BGC0003172, BGC0003173, BGC0003174, respectively, and are also available in our Supplemental Dataset at Zenodo. The antiSMASH v8 outputs for the BGCs are also available in the Supplemental Dataset.

The following datasets were generated:

| Author(s) | Year | Dataset title | Dataset URL | Database and Identifier |
|---|---|---|---|---|
| Reitz ZL | 2026 | Supplemental Dataset for Automated genome mining predicts structural diversity and taxonomic distribution of peptide metallophores across bacteria | https://doi.org/10.5281/zenodo.18866949 | Zenodo, 10.5281/zenodo.18866949 |
| Reitz ZL | 2026 | Enterobactin biosynthetic gene cluster from Buttiauxella brennerae DSM 9396 | https://mibig.secondarymetabolites.org/go/BGC0003172 | MIGBiG, BGC0003172 |
| Reitz ZL | 2026 | Marinobactin biosynthetic gene cluster from Terasakiispira papahanaumokuakeensis DSM 29361 | https://mibig.secondarymetabolites.org/go/BGC0003173 | MIGBiG, BGC0003173 |
| Reitz ZL | 2026 | Ornicorrugatin biosynthetic gene cluster from Pseudomonas brassicacearum DSM 13227 | https://mibig.secondarymetabolites.org/go/BGC0003174 | MIGBiG, BGC0003174 |

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
