## [Editor Report · eLife Assessment]

This **important** and **compelling** study establishes a robust computational and experimental framework for the large-scale identification of metallophore biosynthetic clusters. The work advances beyond current standards, providing theoretical and practical value across microbiology, bioinformatics, and evolutionary biology.

---

## [Referee Report · Reviewer #1 (Public review)]

This work by Reitz, Z. L. et al. developed an automated tool for high-throughput identification of microbial metallophore biosynthetic gene clusters (BGCs) by integrating knowledge of chelating moiety diversity and transporter gene families. The study aimed to create a comprehensive detection system combining chelator-based and transporter-based identification strategies, validate the tool through large-scale genomic mining, and investigate the evolutionary history of metallophore biosynthesis across bacteria.

Major strengths include providing the first automated, high-throughput tool for metallophore BGC identification, representing a significant advancement over manual curation approaches. The ensemble strategy effectively combines complementary detection methods, and experimental validation using HPLC-HRMS strengthens confidence in computational predictions. The work pioneers global analysis of metallophore diversity across the bacterial kingdom and provides a valuable dataset for future computational modeling.

Some limitations merit consideration. First, ground truth datasets derived from manual curation may introduce selection bias toward well-characterized systems, potentially affecting performance assessment accuracy. Second, the model's dependence on known chelating moieties and transporter families constrains its ability to detect novel metallophore architectures, limiting discovery potential in metagenomic datasets. Third, while the proposed evolutionary hypothesis is internally consistent, it lacks further validation.

The authors successfully achieved their stated objectives. The tool demonstrates robust performance metrics and practical utility through large-scale application to representative genomes. Results strongly support their conclusions through rigorous validation, including experimental confirmation of predicted metallophores via HPLC-HRMS analysis.

The work provides significant and immediate impact by enabling transition from labor-intensive manual approaches to automated screening. The comprehensive phylogenetic framework advances understanding of bacterial metal acquisition evolution, informing future studies on microbial metal homeostasis. Community utility is substantial, since the tool and accompanying dataset create essential resources for comparative genomics, algorithm development, and targeted experimental validation of novel metallophores.

Comments on revisions:

I am satisfied with the revisions made by the authors, and they have adequately addressed the concerns raised in the previous version of the manuscript.

---

## [Referee Report · Reviewer #2 (Public review)]

Summary:

This study presents a systematic and well-executed effort to identify and classify bacterial NRP metallophores. The authors curate key chelator biosynthetic genes from previously characterized NRP-metallophore biosynthetic gene clusters (BGCs) and translate these features into an HMM-based detection module integrated within the antiSMASH platform.

The new algorithm is compared with a transporter-based siderophore prediction approach, demonstrating improved precision and recall. The authors further apply the algorithm to large-scale bacterial genome mining and, through reconciliation of chelator biosynthetic gene trees with the GTDB species tree using eMPRess, infer that several chelating groups may have originated prior to the Great Oxidation Event.

Overall, this work provides a valuable computational framework that will greatly assist future in silico screening and preliminary identification of metallophore-related BGCs across bacterial taxa.

Strengths:

(1) The study provides a comprehensive curation of chelator biosynthetic genes involved in NRP-metallophore biosynthesis and translates this knowledge into an HMM-based detection algorithm, which will be highly useful for the initial screening and annotation of metallophore-related BGCs within antiSMASH.

(2) The genome-wide survey across a large bacterial dataset offers an informative and quantitative overview of the taxonomic distribution of NRP-metallophore biosynthetic chelator groups, thereby expanding our understanding of their phylogenetic prevalence.

(3) The comparative evolutionary analysis, linking chelator biosynthetic genes to bacterial phylogeny, provides an interesting and valuable perspective on the potential origin and diversification of NRP-metallophore chelating groups.

Weaknesses:

(1) Although the rule-based HMM detection performs well in identifying major categories of NRP-metallophore biosynthetic modules, it currently lacks the resolution to discriminate between fine-scale structural or biochemical variations among different metallophore types.

(2) While the comparison with the transporter-based siderophore prediction approach is convincing overall, more information about the dataset balance and composition would be appreciated. In particular, specifying the BGC identities, source organisms, and Gram-positive versus Gram-negative classification would improve transparency. In the supplementary tables, the "Just TonB" section seems to include only BGCs from Gram-negative bacteria-if so, this should be clearly stated, as Gram type strongly influences siderophore transport systems.

Comments on revisions:

The authors have adequately addressed all of my previous comments. I have no further comments on the revised manuscript.

---

## [Author Response]

The following is the authors’ response to the original reviews.

**Public Reviews:**

**Reviewer #1 (Public review):**
This work by Reitz, Z. L. et al. developed an automated tool for high-throughput identification of microbial metallophore biosynthetic gene clusters (BGCs) by integrating knowledge of chelating moiety diversity and transporter gene families. The study aimed to create a comprehensive detection system combining chelator-based and transporter-based identification strategies, validate the tool through large-scale genomic mining, and investigate the evolutionary history of metallophore biosynthesis across bacteria.Major strengths include providing the first automated, high-throughput tool for metallophore BGC identification, representing a significant advancement over manual curation approaches. The ensemble strategy effectively combines complementary detection methods, and experimental validation using HPLC-HRMS strengthens confidence in computational predictions. The work pioneers a global analysis of metallophore diversity across the bacterial kingdom and provides a valuable dataset for future computational modeling.Some limitations merit consideration. First, ground truth datasets derived from manual curation may introduce selection bias toward well-characterized systems, potentially affecting performance assessment accuracy. Second, the model's dependence on known chelating moieties and transporter families constrains its ability to detect novel metallophore architectures, limiting discovery potential in metagenomic datasets. Third, while the proposed evolutionary hypothesis is internally consistent, it lacks direct validation and remains speculative without additional phylogenetic studies.The authors successfully achieved their stated objectives. The tool demonstrates robust performance metrics and practical utility through large-scale application to representative genomes. Results strongly support their conclusions through rigorous validation, including experimental confirmation of predicted metallophores via HPLC-HRMS analysis.The work provides a significant and immediate impact by enabling the transition from labor-intensive manual approaches to automated screening. The comprehensive phylogenetic framework advances understanding of bacterial metal acquisition evolution, informing future studies on microbial metal homeostasis. Community utility is substantial, since the tool and accompanying dataset create essential resources for comparative genomics, algorithm development, and targeted experimental validation of novel metallophores.

We thank the reviewer for their valuable feedback. We appreciate the positive words, and agree with their listed limitations. Regarding the following comment:

“Third, while the proposed evolutionary hypothesis is internally consistent, it lacks direct validation and remains speculative without additional phylogenetic studies.”

We agree that additional phylogenetic analyses are needed in future studies. For the revised manuscript, we have validated our evolutionary hypotheses by additionally analyzing two gene families using the likelihood-based tool AleRax, which implements a probabilistic DTL model. The results were consistent with the eMPRess parsimony-based reconstructions, showing comparable patterns of rare duplication, moderate gene loss, and extensive horizontal transfer. Both methods identified similar lineages as the most probable origin and major recipients of transfer events. This agreement between independent reconciliation frameworks supports the reliability of our evolutionary conclusions. We have added a statement referencing this cross-method validation in the revised manuscript.

**Reviewer #2 (Public review):**
Summary:This study presents a systematic and well-executed effort to identify and classify bacterial NRP metallophores. The authors curate key chelator biosynthetic genes from previously characterized NRP-metallophore biosynthetic gene clusters (BGCs) and translate these features into an HMM-based detection module integrated within the antiSMASH platform.The new algorithm is compared with a transporter-based siderophore prediction approach, demonstrating improved precision and recall. The authors further apply the algorithm to large-scale bacterial genome mining and, through reconciliation of chelator biosynthetic gene trees with the GTDB species tree using eMPRess, infer that several chelating groups may have originated prior to the Great Oxidation Event.Overall, this work provides a valuable computational framework that will greatly assist future in silico screening and preliminary identification of metallophore-related BGCs across bacterial taxa.Strengths:(1) The study provides a comprehensive curation of chelator biosynthetic genes involved in NRP-metallophore biosynthesis and translates this knowledge into an HMM-based detection algorithm, which will be highly useful for the initial screening and annotation of metallophore-related BGCs within antiSMASH.(2) The genome-wide survey across a large bacterial dataset offers an informative and quantitative overview of the taxonomic distribution of NRP-metallophore biosynthetic chelator groups, thereby expanding our understanding of their phylogenetic prevalence.(3) The comparative evolutionary analysis, linking chelator biosynthetic genes to bacterial phylogeny, provides an interesting and valuable perspective on the potential origin and diversification of NRP-metallophore chelating groups.

We greatly appreciate these comments.

Weaknesses:(1) Although the rule-based HMM detection performs well in identifying major categories of NRP-metallophore biosynthetic modules, it currently lacks the resolution to discriminate between fine-scale structural or biochemical variations among different metallophore types.

We agree that this is a current limitation to the methodology. More specific metallophore structural prediction is among our future goals for antiSMASH. We have added a statement to this effect in the conclusion.

(2) While the comparison with the transporter-based siderophore prediction approach is convincing overall, more information about the dataset balance and composition would be appreciated. In particular, specifying the BGC identities, source organisms, and Gram-positive versus Gram-negative classification would improve transparency. In the supplementary tables, the "Just TonB" section seems to include only BGCs from Gram-negative bacteria - if so, this should be clearly stated, as Gram type strongly influences siderophore transport systems.

The reviewer raises good points here. An additional ZIP file containing all BGCs used for the manual curation was inadvertently left out of the supplemental dataset for the first version of the manuscript. We have added columns with source organisms and Gram stain (retrieved from Bacdive) to Table S2. F1 scores were similar for Gram positive and negative subsets, as seen in the new Table S2.

We thank the reviewer for suggesting this additional analysis, and have added a brief statement in the revised manuscript.

The “Just TonB” section (in which we tested the performance of requiring TonB without another transporter) was not used for the manuscript. We will preserve it in the revised Table S2 for transparency.

**Recommendations for the authors:**

**Reviewer #1 (Recommendations for the authors):**
(1) In line 43:"excreted" should be replace by "secreted".

Done.

(2) In lines 158-159:"we manually predicted metallophore production among a large set of BGCs."If they are first "annotated with default antiSMASH v6.1", then it is not entirely manual, right? I would suggest making this sentence clearer.

We have revised the language.

(3) In lines 165-169:It would be good to show the confusion matrix of these results.

The confusion matrices are found in Table S2, columns AL-AR.

(4) In Table 1:Method names (AntiSMASH rules/Transporter genes) could be misleading, since they are all AntiSMASH-based, right?

We have adjusted the methods to clarify that while the transporter genes were detected using a modified version of antiSMASH, they are not related to our chelator-based detection rule (which is now correctly singular throughout the text).

(5) Line 198:There are accidental spaces and characters inserted here.

We could not find any accidental spaces and characters here.

(6) Line 209:"In total, 3,264 NRP metallophore BGC regions were detected"Is this number correct? I don't see a correspondence in Table 1.

We have added the following sentence to the Table 1 legend: “An additional 54 BGC regions were detected as NRP metallophores without meeting the requirements for the antiSMASH NRPS rule.”

(7) Line 294:"From B. brennerae, we identified four catecholic compounds"From the bacterial cells or the culture supernatant? I think it is important to state this in a more precise way. If it is from the supernatant, it could be from EVs.

We state in line 292 that “organic compounds were extracted from the culture supernatants”. As our goal was only to confirm the ability of the strains to produce the predicted metallophores, the precise localization (including cell pellet or EVs) was not explored.

(8) Lines 349-357:These results would benefit greatly from a visualization strategy.

Thank you, we have added a reference to the existing visualization in Fig. 5, Ring C.

(9) Lines 452-454:How could clusters be de-replicated? Is there an identity equivalence scheme or similarity metric?

The BGC regions were de-replicated with BiG-SCAPE, which uses multiple similarity metrics as described in Navarro-Muñoz et al, 2020. Clusters could be dereplicated further using a more strict cutoff.

(10) Line 457:"relatively low number of published genomes."Could metagenome-assembled genomes help in that matter?

This is a good question, but we find that MAGs are usually too fragmented to yield complete NRPS BGC regions. We’ve added additional sentences earlier in the discussion: “Detection rates were also lower for fragmented genomes; unfortunately, this limitation (inherent to antiSMASH itself) may hinder the identification of metallophore biosynthesis in metagenomes. As long-read sequencing of metagenomes becomes more common, we expect that detection will improve.”

(11) Lines 514-515:"Adequately-performing pHMMs for Asp and His β-hydroxylase subtypes could not be constructed using the above method."What is the overall impact of this discrepancy in the methodology for these specific groups?

The phylogeny-based methodology was used to reduce false positives. We expect this method will have improved precision at the possible expense of recall.

(12) Lines 543-545:"RefSeq representative bacterial genomes were dereplicated at the genus level using R, randomly selecting one genome for each of the 330 genera determined by GTDB"Isn't it more of a random sampling than a dereplication? Dereplication would involve methods such as ANI computation.

You are correct; we have adjusted the language to clarify.

(13) Lines 559-560: "were filtered to remove clusters on contig edges."This sentence is confusing because networks will be mentioned soon, and they also have edges (not the edges mentioned here), and they could also be clustered (not the clusters mentioned here). Is there a way to make the terminology clearer?

Thank you, we have adjusted the text to read “BGC regions on contig boundaries”

(14) Line 560:"The resulting 2,523 BGC regions, as well as 78 previously reported BGCs "How many were there before filtering?

We have added the number: 3,264

(15) Lines 579-580:Confusing terminology, as mentioned in Lines 559-560.

Adjusted as above.

General comments and questions:An objective suggestion to enrich the discussion is to address the role of bacterial extracellular vesicles (EVs) as metallophore carriers. Studies show that EVs, such as outer membrane vesicles, can transport siderophores or other metallophores for iron acquisition in various bacteria, functioning as "public goods" for community-wide nutrient sharing. Highlighting this mechanism would add ecological and functional context to the manuscript. In the future, EV-associated metallophore transport could also be considered for integration into computational detection tools.

We thank the reviewer for the suggestion; however, we do not think that such a discussion is needed. We briefly discuss the ecological function of metallophores as public goods (and public bads) in the first paragraph of the introduction. We did not find any reports that EV-associated genes co-localize with metallophore BGCs, which would be required for their presence to be a useful marker of metallophore production.

Is there a feasible path to more generalizable detection of chelating motifs using chemistry-aware features? For example, a machine learning classifier trained on submolecular descriptors (e.g., functional groups, coordination motifs, SMARTS patterns, graph fingerprints, metal-binding propensity scores) could complement the current genome-based approach and broaden coverage beyond known metallophore families. While the discussion mentions future extensions centered on genomic features, integrating chemical information from predicted or known products (or biosynthetic logic inferred from BGC composition) could be explored. A hybrid framework-linking BGC-derived features with chemistry-derived features-may improve both recall for novel metallophore classes and precision in distinguishing true chelators from confounders, thereby increasing overall accuracy.

We can envision a classifier that uses submolecular descriptors to predict the ability of a molecule to bind metal ions. However, starting with a BGC and accurately predicting the structure of a hitherto unknown chelating moiety will likely prove difficult. We have added a sentence to the discussion stating that a future tool could use accessory genes to more completely predict chemical structure.

Although the initial analysis was conducted using RefSeq genomes, what are the anticipated challenges and limitations when scaling this method for BGC prospecting in metagenome-assembled genomes (MAGs), particularly considering the inherent quality differences, assembly fragmentation, and taxonomic uncertainties that characterize MAG datasets compared to curated reference genomes?

Please see our response to comment 10, line 457. Our pHMM-based approach is designed to be robust to organism taxonomy; however, fragmentation is a significant barrier to accurate antiSMASH-based BGC detection (including in contig-level single-isolate genomes, see Table 1).

**Reviewer #2 (Recommendations for the authors):**
(1) In the "Chemical identification of genome-predicted siderophores across taxa" section, it would be helpful to annotate the cross-species similarities between predicted metallophore BGCs and their reference clusters (Ref BGCs). As currently described, the main text seems to highlight the cross-species resolving power of BiG-SCAPE itself rather than demonstrating the taxonomic generalizability of the chelator HMM-based detection module.

Thank you for this comment. We intended to display that the new rule is useful for detecting BGCs in unexplored taxa, but we acknowledge that there is not a great diversity in the strains we selected. We have removed “across taxa” to avoid misleading the reader and clarify our intent.

(2) In addition to using eMPRess for gene-species reconciliation, it may be beneficial to explore or at least reference alternative reconciliation tools to validate the inferred duplication, transfer, and loss (DTL) scenarios. Incorporating such cross-method comparisons would enhance the robustness and credibility of the evolutionary conclusions.

We appreciate this valuable suggestion. To validate the robustness of our reconciliation-based inferences, we additionally analyzed two gene families using the likelihood-based tool AleRax, which implements a probabilistic DTL model. The results were consistent with the eMPRess parsimony-based reconstructions, showing comparable patterns of rare duplication, moderate gene loss, and extensive horizontal transfer. Both methods identified similar lineages as the most probable origin and major recipients of transfer events. This agreement between independent reconciliation frameworks supports the reliability of our evolutionary conclusions. We have added a brief statement referencing this cross-method validation in the revised manuscript.